# Gestational Diabetes in the Arab Gulf Countries: Sitting on a Land-Mine

**DOI:** 10.3390/ijerph17249270

**Published:** 2020-12-11

**Authors:** Mukesh M. Agarwal

**Affiliations:** Department of Pathology, College of Medicine, California University of Science and Medicine, Colton, CA 92324, USA; magarwal7@gmail.com; Tel.: +1-909-566-2618

**Keywords:** gestational diabetes prevalence, diabetes mellitus prevalence, Arab Gulf countries, diabetes mellitus prevention

## Abstract

Type 2 diabetes mellitus (T2DM) has become a modern-day plague by reaching epidemic levels throughout the world. Due to its similar pathogenesis, gestational diabetes (GDM) increases in parallel to T2DM. The prevalence of T2DM (3.9–18.3%) and GDM (5.1–37.7%) in countries of the Arab Gulf are amongst the highest internationally, and they are still rising precipitously. This review traces the reasons among the Arab nations for (a) the surge of T2DM and GDM and (b) the failure to contain it. During the last five decades, the massive oil wealth in many Arab countries has led to the unhealthy lifestyle changes in physical activity and diet. The excess consumption of calories turned the advantageous genes, originally selected for the famine-like conditions, detrimental: fueling obesity and insulin resistance. Despite genetic differences in these populations, GDM—a marker for future obesity and T2DM—can overcome this scourge of T2DM through active follow-up and screening after delivery. However, the health policies of most Arab countries have fallen short. Neglecting this unique chance will miss an irreplaceable opportunity to turn the tide of the T2DM and obesity epidemic in the Middle Eastern Arab Gulf countries—as well as globally.

## 1. Introduction

Hyperglycemia in pregnancy (HIP) is the most common metabolic abnormality during pregnancy. It is further sub-classified by the World Health Organization (WHO) into diabetes in pregnancy (DIP) and gestational diabetes mellitus (GDM). GDM accounts for most patients of HIP [1]. Being common, all aspects GDM are important: screening, diagnosis, treatment, follow-up and prevention. However, in any country, to formulate preventive public health measures, the prevalence and the local risk factors are critical for unraveling the epidemiology.

It is a major challenge to assess the prevalence of GDM accurately; GDM is plagued by a lack of consensus among the different international professional health organizations as they advocate dissimilar algorithms for its screening and diagnosis [2]. Thus, there are major discrepancies in the ability of these criteria to identify women with GDM and their capacity to predict adverse pregnancy outcome [3]. This is because among various health and diabetes associations, the glucose thresholds for the diagnosis of GDM vary even though the same diagnostic oral glucose tolerance test is used (Table 1). Despite repeated appeals for consensus [4], the ideal gold-standard for the diagnosis of GDM continues to be debated. However, although the International Association of Diabetes and Pregnancy Study Groups (IADPSG) guideline has been approved by many global health groups (e.g., WHO, American Diabetes Association (ADA), Australasian Diabetes in Pregnancy Society) different guidelines are being followed within and between hospitals in the same city, county and country [5]. Sometimes, older and obsolete guidelines of these organizations are followed. Thus, apart from prevalence, direct comparisons of GDM burden across countries or regions are challenging given the great heterogeneity in screening approaches, diagnostic criteria, and the population characteristics. 

However, the prevalence of type 2 diabetes mellitus (T2DM) in any population correlates with the prevalence of GDM [6]. Thus, populations with a high prevalence of T2DM also have an increased GDM prevalence and vice versa. In fact, the 1980 and 1985 guidelines of the WHO used the same criteria on the oral glucose tolerance test (OGTT) for the diagnosis of GDM and T2DM; the only difference being the presence of pregnancy. Since the diagnosis of T2DM is more defined and less variable amongst the different organizations, studying T2DM prevalence along with GDM prevalence gives a reasonable estimate of the state of glucose intolerance in any population. In short, though the criteria for the diagnosis for GDM are not consistent amongst the major diabetes and health organizations, they are more uniform for T2DM.

## 2. The Modern World’s Epidemic: Type 2 Diabetes Mellitus

T2DM is an epidemic sweeping across the world [7]. The global prevalence (age-standardized) of diabetes mellitus has nearly doubled since 1980, rising from 4.7 to 8.5% in the adult population [8]. Currently, one in 11 adults has diabetes (415 million) while one out of two (46.5%) adults with diabetes is undiagnosed and one in seven births is affected by gestational diabetes. Every six seconds, one person dies of diabetes-related problems [9]. 

The reasons for this epidemic spiraling exponentially are well established [10]. The Pacific Island of Nauru gives an insight into how a changing lifestyle and consuming extra calories is associated with a precipitous rise in T2DM; the first recorded case of diabetes in Nauru was in 1925, whereas by 1979, 40% of the population suffered from this disease [11]. The population of Nauru was malnourished for ages due to a paucity of food. In the mid-1940s, it was discovered that the soil was rich in phosphates. The unexpectedly and abruptly acquired riches from phosphate mines lead to affluence with increased food consumption; the prevalence of diabetes rose in parallel to the prosperity. There was a massive obesity epidemic with T2DM. In a 2007 WHO report, of the 10 most obese countries in the world, eight were from the Pacific Islands (Nauru, Cook Islands, Tonga, Samoa). A decade ago, Nauru was quoted often as it had the highest prevalence of T2DM in the world; again, it must be stressed that it still rates among countries in the world with the highest prevalence of T2DM. 

Further insight is provided by the investigations on the Pima Indians living in the United States, which helped to comprehend the pathogenesis of T2DM; the quick rise in the changing lifestyles were the main contributors, much akin to the local populations of Nauru. The same story is replaying in many countries acquiring sudden wealth. In 1980, only 1% of Chinese adults had T2DM which skyrocketed to 10% by 2008 [12]. The precipitous increase in T2DM in diverse populations, like the Pima Indians, indigenous populations of Nauru and Chinese adults, are similar in one aspect: obesity was fueled by lifestyle changes including decreased physical activity, and increased calorie intake. The indigenous peoples’ story replaying currently in non-indigenous inhabitants of most countries worldwide has been called a global déjà vu [12]. 

## 3. The Relationship between GDM and Type 2 Diabetes

GDM and T2DM are closely related with a similar pathogenesis and etiology. Originally, GDM was defined to be predictive of T2DM in pregnant women after delivery, though currently GDM is being used to prevent maternal and fetal complications at childbirth through treatment. Once diagnosed with GDM, a woman has a substantial chance of developing T2DM following delivery, with a 5-year cumulative incidence rate of over 50% [13]. As childbearing women are relatively young (often between 18–35 years old), women with GDM developing T2DM get it at a younger age, and the longer duration of diabetes increases their risk of developing its myriad complications [14]. GDM and T2DM share many risk factors (increased body mass index, glucose intolerance in relatives, and belonging to certain racial groups). Obesity, due to insulin resistance, remains a strong risk factor for both T2DM and GDM—and insulin resistance is common to the pathophysiology of both these disorders. They also share many associated genes [15]. A ten-year review (1996–2006) of major population-based and hospital-based studies in different countries confirmed that the prevalence of GDM in any population reflects the T2DM prevalence in that population [14]. Thus, it is no wonder that the prevalence of GDM and T2DM run in parallel [6,14].

## 4. Diabetes in the Arab World

Arab countries share a common language, culture, and religion, but there is a big difference between many of them in terms of economic strength, political environment, and social conditions, e.g., warfare. Much information about the countries of the Arabian Gulf is available from the regional offices of the WHO and The International Diabetes Federation (IDF). Cairo is one of the WHO’s six regional offices around the world. IDF confirms that the Middle East, North Africa, and the Horn of Africa (MENA) Region include 21 countries and territories, and its membership includes 29 local diabetes organizations within these countries [16,17]. The MENA region includes countries whose incomes are: (a) high (e.g., Saudi Arabia, Qatar, United Arab Emirates, Saudi Arabia, Oman, Kuwait); (b) average (e.g., Tunisia, Egypt, Syria, Lebanon, Morocco); or (c) low (e.g., Sudan, Somalia).

It is important to note that in 2019 approximately 55.0 million adults aged 20–79 years were living with T2DM in the MENA region—a population of nearly 583 million people. It is distressing that nearly half (45.0%) of the individuals with T2DM in this region were undiagnosed. One in nine live births were affected by HIP. In 2019, USD 25.0 billion in the MENA region was spent taking care of patients with diabetes [16]. 

Diabetes and its related disorders are important to this region as the prevalence of T2DM in the six richest countries (Kuwait, Oman, Qatar, Saudi Arabia, Bahrain, and the UAE) of this region is among the highest in the world [16]. A review of 77 studies showed that the MENA region had the highest prevalence of GDM with a median estimate of 12.9% (range 8.4–24.5%), followed by Southeast Asia, Western Pacific, South and Central America, Africa, and North America and the Caribbean (median prevalence 11.7, 11.7, 11.2, 8.9, and 7.0%, respectively); Europe had the lowest prevalence [18]. 

## 5. Reason for High Prevalence of T2DM and GDM in Specific Arab Countries

The prevalence of GDM varies from 5.1% in Yemen to 24.9% in the United Arab Emirates (UAE). As pointed out, the reasons for the wide variation are multifold: dissimilar criteria used for diagnosis, often the small cohorts to assess prevalence, and sometimes the poorly designed studies. There are other challenges in this region. Due to the political problems of some countries, almost no data are available. It is not a coincidence that the highest-income countries in the Middle Eastern region i.e., Saudi Arabia, UAE, Kuwait and Qatar have the highest prevalence of T2DM and GDM [19]. Finding oil and gas led to a sudden economic growth making them some of the richest countries in the world. The changes in lifestyle, due to the sudden prosperity, contributed to a marked rise in obesity [20]. The wealth generated by oil-rich resources in countries of the Arabian Gulf have led to better living standards, increased urbanization, increased calorie intake, as well as decreased exercise due to reliance on cars and migrant workers. In short, the increase in T2DM in the Arabic-speaking countries has paralleled the increased economic wealth much akin to the Pacific island, Nauru.

Over the last six decades, though many Arab Gulf countries have expanded economically, other Arab countries have regressed economically due major political upheaval and conflicts, e.g., Yemen. These counties at the other end of the spectrum having a low GDP like Somalia also have a low prevalence of T2DM [21,22].

## 6. Genetic Associations of GDM and T2DM as a Cause of Obesity amongst Arab Countries 

Genetics plays a role in GDM with the stress of pregnancy unmasking glucose intolerance in genetically predisposed women. A review of 22 studies showed that many genetic variants associated with GDM share common susceptibility loci with T2DM [23]. Thus, GDM, like T2DM, is a multigenic disease interacting with the environment.

In the early sixties, the thrifty gene hypothesis postulated that T2DM and obesity were linked due to evolution and genes. In impoverished societies, genes are selected so that in times of plentiful nourishment, they store energy so that can be made available in times of famine. However, this hypothesis was unable to explain the heterogeneity of diabetes and obesity between and within populations; therefore, alternate genetic hypotheses were speculated. Thirty years later, the thrifty phenotype (or fetal origin) hypothesis was offered: undernutrition in fetal life caused poor adaptive responses. Another hypothesis, different from the thrifty and thrifty phenotype hypothesis, followed: the drifty gene hypothesis venturing that the prevalence of thrifty genes (or fat storage genes) is due to a genetic drift (hence the name, drifty) in the genes rather than positive selection. None of these hypotheses, though logical, explain the complete epidemiology for the current epidemic of T2DM, GDM and obesity. However, they are the best explanations that are available and are widely accepted [24]. Therefore, the logic of the thrifty hypothesis is that it was beneficial for the survival of a population in adverse conditions. However, these genes become disadvantageous as famines and food shortage are rare. Increasing affluence, like many Arab countries, led to a rapid rise in the prevalence of obesity with its associated T2DM and GDM.

Another important factor that may have contributed to the increased prevalence of T2DM in the Arabic-speaking countries is the highly prevalent consanguinity, which could help to cluster the genes shared between GDM and T2DM [25]. Since specific susceptibility genes, like those related to insulin secretion and resistance, are seen in certain ethnicities, a genetic testing procedure may be developed around Arab ethnicity to predict GDM and T2DM risk in the future.

## 7. GDM in Specific Arab Countries

The prevalence of GDM in specific countries is discussed in order of their prevalence of diabetes mellitus as listed by the latest edition of the International Diabetes Federation. Diabetes Atlas [26] (Table 2).

### 7.1. Saudi Arabia

Saudi Arabia is in the top ten countries in the world with the highest prevalence of T2DM at 18.3%; it is slated to rise precipitously by 2045 [26]. However, in a study on 17,232 Saudi subjects between 1995 and 2000, the prevalence of T2DM was stated to be higher at 23.7% [27]. In another cohort of 9723 women recruited between 2013 and 2015, 24.2% had GDM, while 4.3% had T2DM predating pregnancy, and the rest were healthy [28]. Moreover, this study found that half the women realized that they suffered from T2DM only after becoming pregnant and the authors stressed that the high GDM prevalence just reflected the high prevalence of T2DM in the general population—as alluded to in this review. The burden of obesity was very high in the whole cohort and especially among women with T2DM. Obesity is rampant in Saudi Arabia, like in other oil-rich Gulf countries. In this study, 65% of the women without diabetes and 82% women with GDM were obese or overweight. In summary, based on these studies, nearly one of three pregnant women in Saudi Arabia are affected by HIP. In Saudi Arabia, like the rest of the Arab countries, the following measures are needed according to the studies from the region: (i) (a) a national program for the screening and management of GDM during and after delivery to standardize care; (b) preconception care must be included in the health care of women with diabetes, which is not being currently done; and (c) a national registry for documenting the adverse effects of GDM. Another study from Saudi Arabia studied the poor dietary awareness amongst women with GDM. They recommend the need for dietary counselling due to the ignorance about a healthy diet in GDM in this country [29].

Studies in Saudi Arabia stress the lack of knowledge of GDM among women. In 9002 adult females, using 12 questions for GDM, a study showed that overall, the knowledge of pregnant women about GDM was poor. However, women (a) in urban areas, (b) working in the medical establishment, and (c) who were educated, were better informed [30]. Other studies have confirmed a similar limited knowledge of GDM stressing the need for education. Telemonitoring in Saudi Arabia can facilitate the close monitoring of women with GDM and motivate patients to adopt a healthy lifestyle [31]. All these measures advised by these studies apply to most countries in the Arab world, where less data are available.

### 7.2. Sudan

In Sudan, studies investigating GDM are scarce. In 2012, a study on 100 pregnant women found that the prevalence of GDM was 2% [32]. This is because GDM was defined as diabetes mellitus in pregnancy by the Clinical Practice Guidelines and Standards of Care of Diabetes Mellitus in Sudan as late as 2011. This clearly shows that the guidelines in Sudan had been failing to incorporate the latest research in GDM. This study seems to point that the data are unreliable, possibly due to errors in data collection, laboratory testing and interpretation. Based on the prevalence of T2DM in similar populations, the prevalence in this study is at least nine times lower. The only available report in the literature regarding risk factors for GDM in Sudanese pregnant women was conducted by Khattab and his colleagues in 2007. In 60 women with GDM, the risk factors (compared to controls) were age ≥30 years, BMI ≥ 25 kg/m^2^, family history of diabetes mellitus, glucosuria and proteinuria [33]. Another study on GDM women in Sudan stressed that these women indulged in a diet rich in cereals and sugars, with a low intake of vegetables. In short, their diet was unhealthy, stressing the need for more education [34]. Sudan exemplifies the problems of data on GDM in the Arab world. Despite fine medical schools, a reasonably well developed health care system, there are few data available. The conclusions from the few studies on prevalence of GDM point to unreliable data.

### 7.3. Bahrain

A 2012 study from Bahrain on 49,552 pregnant women showed that 4982 (10.1%) had GDM. There was an increase in the incidence of gestational diabetes from 7.2% in 2002 to 12.5% in 2010. Maternal age, and weight at the onset of pregnancy and at time of delivery were positively associated with risk of GDM with statistically significant odds ratio [35]. A more recent study on 10,495 pregnant women found the prevalence to be 13.5%; all women were screened for GDM during the 24th to 28th weeks of gestation [36]. The authors comment that this study has shown that the native population of Bahrain is a high-risk ethnic group directly related to the high incidence of T2DM and other risk factors for GDM. It is worth observing that if the more current IADPSG/WHO-2013 criteria were used, which give a higher prevalence, the prevalence of GDM would be even higher in Bahrain. Then, it would potentially parallel the neighboring Saudi Arabia. Long-term measures are essential to prevent T2DM: educate the population about lifestyle and diet, as well as motivate these women with GDM to attend follow-ups and screen for T2DM. 

### 7.4. Qatar

Qatar is another affluent country in the Arabian Gulf with a high prevalence of obesity and T2DM. It has rapidly urbanized in recent decades. Most of the expatriates to Qatar come from countries with a high prevalence of T2DM, i.e., the nearby countries of the Middle East, North Africa, as well as the Indian sub-continent. Due to this high prevalence of diabetes and obesity, Qatar has implemented a universal screening approach for all pregnancies. It has also adopted the WHO-2013/IADPSG criteria. In Qatar, all pregnant women are screened in the first antenatal care visit using fasting blood glucose and HBA1c- to rule out preexisting diabetes. Then, 75 g OGTT is performed between 24 and 32 weeks gestation in low-risk patients and between 16 and 20 weeks gestation in high-risk patients (obese, polycystic ovary syndrome (PCOS), previous history of GDM, previous history of intra-uterine fetal death/macrosomia). The Qatar Stepwise report has shown that the prevalence of diabetes in Qatar was 15.5% and that 70.1% of the population in Qatar are either overweight or obese [37]. Again, this reiterates the problems facing the affluent countries of this region.

In 2016, in a retrospective study of 2000 women, the HIP prevalence was 24.0% of which T2DM was 2.5% and GDM was 21.5% (95% CI 19.7–23.3) [38]. The prevalence was similar in the expatriate and Qatari population. In 2011, a study on 2056 pregnant women showed the prevalence of GDM to be 16.3%. Women with GDM were older with a family history of diabetes increased parity and obesity [39]. The varying diagnostic criteria were responsible for these differences [3]; the earlier study used selective screening (clinical and 50 g glucose challenge test), while the latter used universal screening with the 75 g OGTT using the WHO-2013 criteria, which is known to give a higher prevalence. Again, this shows how the criteria for diagnosis affect the prevalence of GDM in the same country, as alluded to earlier. In short, the prevalence of GDM in Qatar parallels the other rich GCC Arab countries. However, Qatar represents one country where the practices for GDM are comparable to the best in the world. In addition to its small population, the intent of the health planners has gone a long way in achieving this practice.

### 7.5. United Arab Emirates (UAE)

The UAE is a multi-ethnic community where during the last decade, the prevalence of T2DM was repeatedly stated to be the second highest in the world after Nauru [40]. GDM, obesity and multiparity are also very common in the UAE [41]. 

Multiple studies from the UAE exemplify the problems of trying to assess the prevalence of GDM—and the same story is retold in other rich Arab Gulf countries. As there is no national guideline, hospitals in UAE utilize various criteria for GDM diagnosis for multiple reasons. The care-giving physicians often follow guidelines of their institutions like ADA and WHO. The disparity in the background and training of the multi-national obstetricians and endocrinologists (from Canada, India, Australia, UK, Egypt, and USA) makes them use one of the guidelines popular in their own country. Additionally, some hospitals use the older guidelines, which have not been updated just as a matter of convenience [41]. Thus, the prevalence of GDM in the UAE varies from 7.9 to 37.7%, depending on the criteria used for the diagnosis [41,42]. The estimates of T2DM in UAE are 18–20%, which is a reasonable parameter for the prevalence of GDM [43]. Glucose intolerance in the young is important as some of these adolescent girls are prone to develop GDM, if they become pregnant; others may suffer fromT2DM when they reach adulthood. This is a newer problem and an emergence of T2DM in the pediatric population of United Arab Emirates has also been documented [44].

### 7.6. Egypt

Egypt, like most countries, shows a huge variation in the reported prevalence of GDM. In 250 pregnant women, who attended a rural family health center in Egypt, the prevalence of GDM was found to be 8% [45]. A meta-analysis of 33 studies from the MENA region found the overall prevalence of GDM in the entire region to be 11.7%; 887,166 participants were included in this meta-analysis from 2000 to 2018. The results also showed that the highest prevalence of GDM was in Egypt i.e., 24.2%, which the authors comment is due to the methodological differences between the studies included or due to factors such as the older age and higher BMI in these countries [46]. 

According to the IDF, Egypt is among the world’s top 10 countries in the total number of patients with diabetes. The prevalence of T2DM is around 15.2% among adults between 20 and 79 years of age [26], which is a reasonable estimate of the prevalence of GDM in Egypt like in all countries [14]. The prevalence of T2DM in Egypt has almost tripled over the last two decades. This sharp rise could be attributed to either an increased pattern of the traditional risk factors for T2DM such as obesity and physical inactivity as well as a change in the eating patterns or other risk factors unique to Egypt [47].

### 7.7. Lebanon

The prevalence of GDM in Lebanon, like many other countries in the region, is not easily discernable; however, using the prevalence of T2DM as a proxy estimate, this provides a reasonable guess of GDM prevalence [14]. According to the current data, the IDF estimates of diabetes mellitus in Lebanon is 12.9% (10.5–16%) or affecting about 500,000 adults between 20 and 79 years of age. The prevalence of T2DM among the 3000 individuals recruited was found to be 11.3% (95% CI 10.2–12.4) [48]. Again, no other studies could be found addressing the situation of GDM in Lebanon.

### 7.8. Syria

There is a paucity of data available from Syria due to the long-drawn political strife. The IDF reports the T2DM prevalence to be 12.3%. In a survey of 1550 Syrian refugees living in Jordan, the prevalence of diabetes mellitus was 5.3% [49]. However, there are many confounding variables using immigrant data, making the reports unreliable. The IDF estimate in Syria of T2DM is 12.3% and likely the best surrogate estimate for the prevalence of GDM in Syria.

### 7.9. Tunisia

In 2014, in a study of 6580 households in Tunisia, the prevalence of T2DM was 15.1% [50], and the national prevalence as estimated by the IDF is 10.2%, including the greatest proportion of undiagnosed diabetes (75%) in the world after Mozambique and Tanzania [26]. Overall, the prevalence of GDM in Tunisia has been increasing. This increase is mainly caused by environmental changes, nutritional excess due to westernization of lifestyle and the growing prevalence of obesity.

In another study on 30 pregnant women, the majority of women had a limited knowledge of gestational diabetes, its risk factors and its consequences. Only 6% of women knew that overweight and a family history of diabetes can be predictive factors of GDM. [51]. Thus, education and well planned strategies are urgently needed to reduce the burden of diabetes as pointed out by this major study [26].

### 7.10. Jordan

There are few studies about GDM from Jordan. In 644 Jordanian women attending the Jordan University Hospital between January 2015 and January 2016, the prevalence of GDM was 13.5%. The risk factors for these women were maternal age, gravidity, parity, maternal pre-pregnancy BMI, maternal BMI at the time of the glucose testing, the presence of acanthosis nigricans, past history of gestational diabetes, and a family history of diabetes mellitus [52]. The underlying prevalence of diabetes in Jordanian adults is 9.9% [26].

### 7.11. Libya

There are few data available about the state of GDM in Libya due to the political situation. The IDF atlas [26] also says that data are not available. The Libyan Medical Journal has almost no articles on GDM that address the prevalence, management and follow-up of GDM. The prevalence of T2DM is 9.7% in Lybia and is the best estimate available for HIP in this war-torn country.

### 7.12. Oman

Oman is following the trend of the Arab world with an increasing prevalence of T2DM, GDM and glucose intolerance. In 2003, the Ministry of Health of Oman reported a consistent rise in GDM. It has been reported of all pregnant women that 11.87% pregnant women are diagnosed with GDM in Oman, and 14.3% of fetal deaths are attributed to GDM [53]. A cross-sectional study covering different regions of Oman showed that the risk factors for GDM in Omani women are age ≥32 years, a family history of diabetes and gestational diabetes ≥5 pregnancies, ≥3 deliveries, marriage at ≤18 years, and height ≤155 cm [54].

Some studies from Oman and Bahrain reported a lower prevalence of GDM. It hovers around 10% in both countries as claimed by one study [55]. However, like all studies, the prevalence varies with the method used for diagnosis.

### 7.13. Iraq

In a 2014 prospective study on 100 pregnant women, the GDM prevalence was 7%. The women with GDM had an increased risk of hypertension in pregnancy, preterm delivery, and cesarean section. Increasing age, a family history of diabetes, increasing parity, and increase in BMI were the risk factors for developing GDM [56]. In a recent 2020 study on 120 women between the ages of 20–45 years, the prevalence of GDM was found to be 13.3% [57]. None of the studies mention the criteria used for the diagnosis of GDM, which is important. Moreover, the number of pregnant women recruited was small. Furthermore, due to the political situation, currently hardly any studies are being performed. Even the latest edition of the IDF states that the data on HIP from Iraq are unavailable [26].

### 7.14. Morocco

In Morocco, the reported GDM prevalence ranges vary like in other Arab countries. In one study on 403 women screened, 33 had gestational diabetes, i.e., the prevalence was 8.2%. Another study reports a prevalence of 10% [58]. Again, the lower prevalence compared to some other Arab countries may reflect the relative lack of affluence in Morocco. However, GDM prevalence reached 23.7% in Marrakech, which may be a function of using the currently popular IADPSG and WHO criteria; they are known to give a much higher prevalence [41]. However, there seems to be an overall temporal increase in the prevalence of GDM in Morocco [59].

### 7.15. Algeria

The prevalence of T2DM in Algeria is 7.2% [25]. One study showed that the prevalence of T2DM in Algeria had increased from 6.8% in 1990 to 12.29% in 2005, however, it may be still higher in certain groups and areas of the country [60]. In 2014, screening for GDM was not performed in Algeria as part of a standard antenatal care. However, with a push from the scientific societies like scientific societies such as the Algerian Society of Diabetology, a gestational diabetes register was created by the Ministry of Health. Thus, Algeria is aligning itself with the world practices of GDM [60].

### 7.16. Palestine

The Palestinian Territories consist of the West Bank and Gaza. The IDF lists the prevalence of T2DM as 6.7% [26]. T2DM is the fourth leading cause of death in this region. One study based on a model reported a prevalence of 15.3% in 2010 and in adult patients aged 20–79 years and predicted it to increase to 20.8% by 2020, and to 23.4% in 2030 [61]. Even though the prevalence of GDM is not easily available, the risk factors for GDM have been studied. These have been identified as obesity before pregnancy, stillbirth, cesarian sections and a family history of diabetes mellitus [62].

### 7.17. Somalia

There is a paucity of data from Somalia. However, data are available on Somali immigrants. In 583 Somali women in Finland, the prevalence of GDM was 14.4% [63]. Data from immigrants have other biases; therefore, it is harder to extrapolate to the situation in Somalia.

### 7.18. Yemen

There are few data available from Yemen. This is understandable due to the political strife and an under resourced health system. Only one study on GDM is available and it was before the current war and pandemic. The prevalence of GDM in a cohort of 311 women was found to be 5.1% among the study population [64]. Multivariate analysis confirmed age ≥30 years, previous GDM, a family history of diabetes, and a history of PCOS as independent risk factors for GDM prevalence. However, the criteria used for GDM were the criteria for T2DM and no OGTT was used. A pregnant woman with GDM was confirmed, if on 2 consecutive days, the fasting capillary glucose or random capillary glucose were ≥126 mg/dL or 200 mg/dL, respectively, which are criteria to make a diagnosis of T2DM. A glucometer was used and not venous plasma glucose. The 2019 edition of the IDF [26] states that the prevalence of diabetes in Yemen is 3.9% (95% CI 3.0–11.1).

## 8. Future Directions for the Arab World

### 8.1. Increasing Awareness among Care-Receivers and Care-Givers

The rampant increase in T2DM and GDM in the Arab countries behooves health planners to aim at prevention. Pursuing women with GDM after delivery can contain the epidemic of T2DM through preventive strategies [65]. Arab countries have few available data on GDM, most information being limited to prevalence and almost no data about follow-up after delivery, as shown by this review and iterated by a study from Saudi Arabia [66]. In general, recommended testing for women with GDM after delivery all over the world is poor at best [67]. In Arab countries, most women with GDM are lost in follow-up after delivery. However, the few available studies stress the failure of follow-up and the lack of knowledge among patients about GDM [29]. Regrettably, there is a paucity of knowledge about GDM even amongst the care-givers. A physician survey assessed the current regional practices of screening, diagnosis, and follow-up of GDM and knowledge of GDM within seven hospitals in UAE and one in Oman. Approximately 40% pf physicians, who were obstetricians and endocrinologists, were not aware of a major study in GDM, i.e., the Hyperglycemia and Adverse Pregnancy Outcome (HAPO) Study or the IADPSG. As this study stresses, more awareness and education of care-givers would help in improving follow-up [68]. It has also been argued that the follow-up of GDM is not the ideal means to prevent T2DM. Preconception programs are also effective [69]; however, the two are not mutually exclusive.

### 8.2. Effects of Migration to the Arab World

Migration to the Arab world is an important factor affecting prevalence and follow-up. The large expatriate Arab, Asian and European migrant populations in Arab countries may influence the reported prevalence. For instance, the large expatriate European populations (with low GDM prevalence) in Qatar may lead to an inaccurate reported prevalence. The opposite is true when Arab women migrate to the western world where the GDM prevalence is lower; thus, they can influence the prevalence of their adapted country [70]. Often, the expatriate population in Arab countries cannot be followed after their delivery, as these women often return permanently to their home country. Health planners need to incorporate the effect of migration by carefully studying their population demographics when they formulate guidelines.

### 8.3. Overcoming Barriers to Follow-Up GDM

As can be appreciated from this review, studies following up women after GDM are almost non-existent. The barriers to follow-up in Arab countries can only be extrapolated from countries with available data [71]. These impediments have been cited as follows: (a) patient lost to follow-up; (b) health care provider (in the obstetric setting) not seeing the patient; (c) patient not considering follow-up tests necessary after delivery; (d) stress of adjusting to a new baby; and (e) feeling healthy post-partum. In the Arab world, the ubiquitous fatalism may be an additional cultural factor. All these aspects will help in planning effective strategies. The similarity in problems affecting many Arab countries calls for some unifying decisions. Updating screening guidelines to the latest research will help to reduce these differences. More education for the patients and the care-givers, especially in rural settings, would be a major step in the right direction.

### 8.4. Adapting International Guidelines Locally

If the local diabetes and health organizations do not offer any guidelines for GDM, help is available from many international organization guidelines. However, many algorithms may prove too costly and thus impossible to follow. The International Federation of Gynecology and Obstetrics (FIGO) represents professional societies in 132 countries and has an excellent approach to GDM that can be varied according to the local situation. Arab countries, without any local guidelines, would be well served to follow the FIGO approach to GDM [72]. It is practical, adaptable, scientific, rational, comprehensive, current, and easily available free of any cost. Essentially, FIGO recommends the WHO-2013 guidelines, which have been adapted by most major organizations in the world. In short, following current guidelines with a better follow-up and routine screenings after delivery, would go a long way in managing women with GDM.

## 9. Conclusions

GDM is not just a harbinger of T2DM after childbirth as we believed in the past; it is a unique chance to identify women and infants at risk for future obesity, T2DM, and cardiovascular disease [73]. Several opportunities are available for the prevention of T2DM in women with a history of GDM, such as education about risk awareness, implementation of a healthy lifestyle, breast-feeding and pharmacotherapy [65]. However, following the delivery of their infant, women with GDM fail to visit physicians and are usually lost to follow-up; however, these women do visit health services focused on the wellbeing of their newly born infant, which is an opportunity to provide them with follow-up advice [71].

Finally, the onus is on health care providers to motivate women with GDM for follow-up after delivery and to periodically check for the impending T2DM while instituting life-style changes [74]. It is imperative to follow-up women with GDM after delivery. If we can achieve this landmark of following women with GDM after childbirth, we shall succeed in “turning the tide” of T2DM epidemic in the Arab Gulf countries of the Middle East—and the world.

## Figures and Tables

**Table 1 ijerph-17-09270-t001:** Diagnostic criteria of major organizations for gestational diabetes mellitus (GDM) with diagnostic values for the 75 g oral glucose tolerance test over time.

Criteria ^#^	Year	Abnormal Glucose Values Needed for Diagnosis	Plasma Glucose Thresholds mmol/L		
			0 h	1 h	2 h
IADPSG	2010	≥1	5.1	10.0	8.5
CDA	2013	≥1	5.3	10.6	9.0
WHO (NICE) *	1999 (2008)	≥1	7.0	-	7.8
ADA (C&C) *	2003 (1982)	≥2	5.3	10.0	8.6
ADIPS	1998	≥1	5.5	-	8.0
NZSSD	2004	≥1	5.5	-	9.0
EASD	1996	≥1	6.0		9.0
CDA	2003	≥2	5.3	10.6	8.9

^#^ IADPSG, International Association of Diabetes and Pregnancy Study Groups; CDA, Canadian Diabetes Association; WHO, World Health Organization; NICE, National Institute for Health and Care Excellence; ADA, American Diabetes Association; C&C, Carpenter and Coustan; ADIPS, Australasian Diabetes in Pregnancy Society; NZSSD, New Zealand Society for the Study of Diabetes; EASD, European Association for the Study of Diabetes. * The diagnostic criteria of these guidelines are the same.

**Table 2 ijerph-17-09270-t002:** Prevalence of type 2 diabetes mellitus (T2DM) and GDM sorted by the prevalence of diabetes mellitus (DM).

Country	Prevalence of GDM (%)	Prevalence of DM (%)
Saudi Arabia	24.2	18.3
Sudan	Unknown	17.9
Bahrain	10.1–13.5	16.3
Qatar	19.7–23.3	15.5
United Arab Emirates	7.9–37.7	15.4
Egypt	8.0–24.2	15.2
Lebanon	Unknown	12.9
Syria	Unknown	12.3
Tunisia	Unknown	10.2
Jordan	13.3	9.9
Libya	Unknown	9.7
Oman	10.5	8.0
Iraq	13.3	7.6
Morocco	8.2–23.7	7.4
Algeria	Unknown	7.2
Palestine	Unknown	6.7
Somalia	14.4	4.8
Yemen	5.1	3.9

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
