# Peer review of "Gestational Diabetes in the Arab Gulf Countries: Sitting on a Land-Mine"

_ijerph, 2020, doi:10.3390/ijerph17249270_

Round 1
Reviewer 1 Report
Thank you for your research contributions to this very important topic. The following comments, questions, and suggested edits are offered with all due respect and for your consideration:
Abstract
Line 18-20: “GDM... can help to thwart this scourge of T2DM through restraint and education...” Suggest rephrasing this for clarity. For example, could intensive education and behavioral intervention help reduce T2DM risk for some women who experience GDM? Will genetics play a role for some women with GDM, whereby they will develop T2DM later even if they exercise “restraint” and receive “education”?
Line 20: "However, Arab countries have largely failed to actively pursue women with GDM...” Suggest rephrasing this for clarity. For example, have women with GDM been lost to followup? Have public health policies and/or standard of care for GDM omitted followup for women with GDM?
1. Introduction
Lines 29-30: "Being common, all aspects GDM, from screening, diagnosis, and treatment to follow-up, are important.” Is there a word missing?
Line 56: "underlying population” It is unclear who this term is referring to.
Lines 56-60: The prevalence of T2DM is used as an estimate of GDM prevalence, but the rationale for could be stronger by providing supporting references. The author may opt to refute suggestions that using T2DM as an estimate for GDM is not reliable (doi: 10.1155/2012/721653).
Line 60, 177, 177, 179, 182: “diabetic” Author may consider the use of the word “diabetes” instead of “diabetic"
2. The modern world’s epidemic: diabetes mellitus
Lines 67-71, 72-88: The author may wish to reorganization these sentences/paragraphs to strengthen the overall impact of the information.
Line 86, 130, 176: The author may wish to exercise care with the use of terms that suggest causation, instead of association (“Thus,” “just reflects").
3. The relationship between GDM and Type 2 Diabetes
Line 95-96: It may help to quantify “young"
4. Diabetes in the Arab World
Lines 107-108, 112-115: Author may wish to reword/rephrase this for clarity.
Line 119: Is “the Middle East and North Africa” referring to MENA as it is in Line 107?
5. Reason for high prevalence of DM and GDM in Arabs
-Author may wish to reconsider subheading for continuity. For example, “T2DM” instead of “DM"
Line 130: Grammar?
Lines 136-137: References 21 and 22 may belong at the end of the sentence?
6. Genetic hypotheses and T2DM
Lines 141 and 142 may contain repetitive information.
Line 147: Is it “drifty” or “thrify”?
Line 150: “(in part of full)”? It may help to rephrase this.
Line 156: Migration is introduced as a factor and later referred to in (e.g., 7.4, 7.8). The author may wish to add information (e.g., reference) that establishes this as a known factors associate with T2D and/or GDM prevalence rates.
Line 165: Should “T2Dm” be “T2DM” for continuity?
7.1 Saudi Arabia
Line 169: For clarification, it may help to specify what the author is referring to when using the term “diabetes.” For example, is this referring to all types of diabetes or to T2DM?
Line 177: It may help to clarify what is meant by “assessing T2DM is easier.” Is this related to guidelines? Available screening? Something else?
Line 183: Incomplete sentence? (“A nationals registry..."
Line 185, 248: Author may consider the use of the word “healthy” instead of “good"
Line 187: With regard to the use of the term “confirmed,” did the authors of the study “confirm" that this cohort of 9,002 adult females indeed had previously suspected limited knowledge about GDM? The author may consider rewording this for clarification.
Line 189: The author may wish to use “limited knowledge” instead of “ignorance"
7.2 Sudan
Line 195: It would seem that the percentage would be higher if GDM was more broadly "defined as diabetes mellitus."
Line 199-200: It may help to qualify what is meant by “the data is unreliable."
7.3 Bahrain
Lines 213-214: What types of "long-term measures” would be helpful? Would this include screenings for T2DM/GDM during initial prenatal visits?
7.4 Qatar
It may help to reorganize this section for clarity. Also, it will help to define acronyms introduced here (ANC, PCOS). Of note, PCOS is defined later (line 341).
Lines 230-231: The author may wish to reorganize this sentence.
Line 234: Is there a reference to support, “The varying diagnosis criteria...”?
7.5 United Arab Emirates (UAE)
Line 237: The author may wish to change “T2 DM” to “T2DM” for continuity.
Line 245, 246: Are reference available to support these statements?
Line 249: The author may wish to change “Type 2 diabetes” to "T2DM” for continuity.
7.6 Egypt
Line 261-262: The author may wish to add a reference or justification for the use of T2DM as a reliable estimate of GDM.
7.7 Lebanon
As previously noted, the author may wish to add a reference or justification for the use of T2DM as a reliable estimate of GDM.
Lines 267-271: The author may wish to reorganize this paragraph for clarity.
Line 270: The author may wish to change “T2 DM” to "T2DM” for continuity.
7.8 Syria
Line 274: Does a reference to “Syrian refugees living in Jordan” introduce the possibility that confounding variables impact this data?
7.9 Tunisia
Line 281: The paragraph begins with the topic of increasing prevalence, but then continues with information about GDM knowledge. It may help to reorganize this paragraph/section for clarity.
7.12 Oman
Mortality and specific risk factors are listed i this section, but not in previous sections. It may help to restructure sections 7.1-7.18 and/or to consider adding a table to streamline the data reported for each country.
7.13 Iraq
The author may wish to reorganize this paragraph for clarity.
7.14 Morocco
Line 316-317: It may help to rephrase these sentences for clarity.
7.15 Algeria
Is data from the “gestational diabetes register” available? If so, the author may want to include it.
7.16 Palestine
Line 328: The author may want to use “T2DM” instead of “DM” for continuity, unless the author is referring to all types of diabetes.
Lines 331-333: It may help to rephrase these sentences for clarity.
7.17 Somalia
It may help to rephrase these sentences for clarity.
7.18 Yemen
Line 343: It may help to rephrase this sentence for clarity.
8. Future Directions for the Arab world
Line 349: The author may wish to use the term “women with GDM” instead of “GDM women"
Lines 353-354: Would the implementation of routine screenings and a consensus about the diagnosis of GDM be beneficial? Could this help with timely diagnosis of T2DM (pre or post pregnancy) and GDM?
9. Conclusions
A few concepts are introduced here (e.g., cardiovascular disease, breastfeeding, pharmacotherapy). It may help to introduce these topics earlier in there manuscript OR to highlight why these topics are important based on the information in the manuscript (i.e., information obtained from the review).
It may help to review Lines 363, 370, 371 for clarification.
Author Response
Thank you for your time and extensive review. It has given our review a new dimension. We are very grateful, indeed.
All our responses, point by point, are in red.
Abstract
Line 18-20: “GDM... can help to thwart this scourge of T2DM through restraint and education...” Suggest rephrasing this for clarity. For example, could intensive education and behavioral intervention help reduce T2DM risk for some women who experience GDM? Will genetics play a role for some women with GDM, whereby they will develop T2DM later even if they exercise “restraint” and receive “education”?
Thank you. We have rephrased the sentence by removing thwart (it is too strong) and adding despite the genetic differences, as you suggest. In fact, we have rewritten major portions of the manuscript to incorporate all your valid concerns and excellent suggestions.
Line 20: "However, Arab countries have largely failed to actively pursue women with GDM...” Suggest rephrasing this for clarity. For example, have women with GDM been lost to followup? Have public health policies and/or standard of care for GDM omitted followup for women with GDM?
We have added public health policies and replaced pursue (after delivery) with follow-up. We have added that the health policies have fallen short. We have rephrased the entire abstract for clarity. We hope you will like the new version.
- Introduction
Lines 29-30: "Being common, all aspects GDM, from screening, diagnosis, and treatment to follow-up, are important.” Is there a word missing?
We have rephrased the sentence for clarity, and it reads much better. Thank you.
Line 56: "underlying population” It is unclear who this term is referring to.
Thank you. Underlying was superfluous and confusing. We have removed it.
Lines 56-60: The prevalence of T2DM is used as an estimate of GDM prevalence, but the rationale for could be stronger by providing supporting references. The author may opt to refute suggestions that using T2DM as an estimate for GDM is not reliable (doi: 10.1155/2012/721653).
Thank you for this reference. We have used it in our references (68) and argue that the two are not mutually exclusive: using GDM and “other methods” to prevent future DM2. We have expanded this important point in future directions section (8.1).
Line 60, 177, 177, 179, 182: “diabetic” Author may consider the use of the word “diabetes” instead of “diabetic"
Thank you. We have changed all the diabetic in the nine places in the document. Only when used in the references, it has been retained.
- The modern world’s epidemic: diabetes mellitus
Lines 67-71, 72-88: The author may wish to reorganization these sentences/paragraphs to strengthen the overall impact of the information.
Thank you. We have reorganized the paragraphs as you suggested. The logical flow is far better now.
Line 86, 130, 176: The author may wish to exercise care with the use of terms that suggest causation, instead of association (“Thus,” “just reflects").
You are right. We were not implying causation, and we have made sure it is not implied by using your suggestions.
- The relationship between GDM and Type 2 Diabetes
Line 95-96: It may help to quantify “young"
We have clarified that young implies women between 18-35 years of age. In fact, they are often younger in the Arab world.
- Diabetes in the Arab World
Lines 107-108, 112-115: Author may wish to reword/rephrase this for clarity.
We have made these changes. Thank you.
Line 119: Is “the Middle East and North Africa” referring to MENA as it is in Line 107?
Yes. We have used the acronym instead of the full form 2 times. We thank you for picking up this error.
- Reason for high prevalence of DM and GDM in Arabs
-Author may wish to reconsider subheading for continuity. For example, “T2DM” instead of “DM"
We have done so. Also, we searched for DM in the document, but there were none. Thank you.
Line 130: Grammar?
We have rephrased the sentence. Thank you.
Lines 136-137: References 21 and 22 may belong at the end of the sentence?
We apologize. We have changed it. Thank you.
- Genetic hypotheses and T2DM
Lines 141 and 142 may contain repetitive information.
We agree completely. We have removed a line. We hope the changes will satisfy you.Thank you.
Line 147: Is it “drifty” or “thrify”?
It is drifty. We agree that it could be confused as the words rhyme. We have clarified that it was a ‘new’ hypothesis.
Line 150: “(in part of full)”? It may help to rephrase this.
We have rephrased. Thank you.
Line 156: Migration is introduced as a factor and later referred to in (e.g., 7.4, 7.8). The author may wish to add information (e.g., reference) that establishes this as a known factors associate with T2D and/or GDM prevalence rates.
This is a very important point. We have added a new para in the future directions for the Arab world (8.2) and added a reference (69) which is a meta-analysis on migration of women with GDM.
Line 165: Should “T2Dm” be “T2DM” for continuity?
That was a typographical error. Thank you.
7.1 Saudi Arabia
Line 169: For clarification, it may help to specify what the author is referring to when using the term “diabetes.” For example, is this referring to all types of diabetes or to T2DM?
We have changed it to DM2. Since over 99% of diabetes is T2DM, we can presume diabetes is T2DM. However, many references just specify diabetes, so it becomes hard to decipher if they mean T2DM or overall diabetes. Overall, we interpret diabetes to be T2DM.
Line 177: It may help to clarify what is meant by “assessing T2DM is easier.” Is this related to guidelines? Available screening? Something else?
We have clarified that ‘assessing T2DM’ refers to ‘assessing T2 DM prevalence. Corrected. Thank you.
Line 183: Incomplete sentence? (“A nationals registry..."
Corrected. We have rephrased the whole sentence.
Line 185, 248: Author may consider the use of the word “healthy” instead of “good"
We have changed the five good parameters in the manuscript. It is much better now. Thank you.
Line 187: With regard to the use of the term “confirmed,” did the authors of the study “confirm" that this cohort of 9,002 adult females indeed had previously suspected limited knowledge about GDM? The author may consider rewording this for clarification.
We have changed it to a milder showed.
Line 189: The author may wish to use “limited knowledge” instead of “ignorance"
Changed. Thank you.
7.2 Sudan
Line 195: It would seem that the percentage would be higher if GDM was more broadly "defined as diabetes mellitus."
We fully agree. This is the reason that when uses 100 women, the results are unreliable. We have clarified this in the manuscript.
Line 199-200: It may help to qualify what is meant by “the data is unreliable."
We have qualified why the data is potentially unreliable. Thank you.
7.3 Bahrain
Lines 213-214: What types of "long-term measures” would be helpful? Would this include screenings for T2DM/GDM during initial prenatal visits?
We have clarified the long-term measures in detail. Thank you.
7.4 Qatar
It may help to reorganize this section for clarity. Also, it will help to define acronyms introduced here (ANC, PCOS). Of note, PCOS is defined later (line 341).
We have spelt out the acronyms. Thank you.
We have clarified PCOS and used the acronym later.
Lines 230-231: The author may wish to reorganize this sentence.
Indeed, this was a badly written sentence. Rephrased. Many thanks.
Line 234: Is there a reference to support, “The varying diagnosis criteria...”?
We have expanded on the criteria used in the two studies and provided a reference.
7.5 United Arab Emirates (UAE)
Line 237: The author may wish to change “T2 DM” to “T2DM” for continuity.
Thank you. Changed.
Line 245, 246: Are reference available to support these statements?
This line is paraphrased from one of our studies and the reference provided.
Line 249: The author may wish to change “Type 2 diabetes” to "T2DM” for continuity.
Thank you. Changed.
7.6 Egypt
Line 261-262: The author may wish to add a reference or justification for the use of T2DM as a reliable estimate of GDM.
We have alluded to this earlier in the paper. We have included the reference. Thank you.
7.7 Lebanon
As previously noted, the author may wish to add a reference or justification for the use of T2DM as a reliable estimate of GDM.
Again, we have alluded to this earlier in the paper. We have added a reference. Thank you.
Lines 267-271: The author may wish to reorganize this paragraph for clarity.
We have made this sentence clearer. Thank you.
Line 270: The author may wish to change “T2 DM” to "T2DM” for continuity.
We have done so. Thank you.
7.8 Syria
Line 274: Does a reference to “Syrian refugees living in Jordan” introduce the possibility that confounding variables impact this data?
Yes, it does. We have added this caveat. Thank you.
7.9 Tunisia
Line 281: The paragraph begins with the topic of increasing prevalence, but then continues with information about GDM knowledge. It may help to reorganize this paragraph/section for clarity.
We have corrected this illogical approach. Now it is much more logical. Thank you.
7.12 Oman
Mortality and specific risk factors are listed i this section, but not in previous sections. It may help to restructure sections 7.1-7.18 and/or to consider adding a table to streamline the data reported for each country.
This is an excellent idea. We had tried to do this earlier. However, almost no data is available from most Arab countries. Oman is an exception due to the medical schools there. UAE and Saudi Arabia have data. Surprisingly, Sudan, which produces fine physicians, barely has even prevalence studies. We have added a new paragraph stressing this in the conclusion. However, we appreciate all your efforts at improving our manuscript. It has made a world of a difference.
7.13 Iraq
The author may wish to reorganize this paragraph for clarity.
We have rewritten portions of the paragraph and transposed sentences.
7.14 Morocco
Line 316-317: It may help to rephrase these sentences for clarity.
We have rewritten the entire section. Thank you.
7.15 Algeria
Is data from the “gestational diabetes register” available? If so, the author may want to include it.
The authors used of the study mention the creation of register. However, it is not available on the internet.
7.16 Palestine
Line 328: The author may want to use “T2DM” instead of “DM” for continuity, unless the author is referring to all types of diabetes.
The paper just mentions diabetes, however, since T2DM makes up over 99% of diabetes, they are interchangeable.
Lines 331-333: It may help to rephrase these sentences for clarity.
We have done so. Thank you.
7.17 Somalia
It may help to rephrase these sentences for clarity.
We have done so. Thank you.
7.18 Yemen
Line 343: It may help to rephrase this sentence for clarity.
We have done so. Thank you.
- Future Directions for the Arab world
Line 349: The author may wish to use the term “women with GDM” instead of “GDM women"
Lines 353-354: Would the implementation of routine screenings and a consensus about the diagnosis of GDM be beneficial? Could this help with timely diagnosis of T2DM (pre or post pregnancy) and GDM?
Yes, of course. We have added this to the future directions. We have expanded on this section. Now, we have four subsections, as this is a very critical part of the review.
- Conclusions
A few concepts are introduced here (e.g., cardiovascular disease, breastfeeding, pharmacotherapy). It may help to introduce these topics earlier in there manuscript OR to highlight why these topics are important based on the information in the manuscript (i.e., information obtained from the review).
It may help to review Lines 363, 370, 371 for clarification.
We have done so. We have worked extensively on the last two headings: Future directions and conclusions. Again, thank you for your time and help.
Reviewer 2 Report
This is a review paper on the overall prevalence of gestational diabetes and type 2 diabetes in the Middle East (Arab) countries.
While the data reported are interesting, several concerns must be addressed before this paper can be accepted for publication-
- It is important to keep the review focused on GDM only- the terms T2DM, DM and GDM have been loosely used in various places of the text with no good integration
- For example, the term 'DM2'- line 84 is misleading-'DM'-line 124- please revise and be consistent
- The content of the review is not well organized- is this simply a review of prevalence data or the causes and potential management strategies?
- subhead number 5- reason for high prevalence of DM and GDM..- this section is very poorly written
- A comparative table on countrywise prevalence and etiologic factors will be very helpful
- Overall, the review contributes little in this area
Author Response
Thank you for your time and review. We are very grateful, indeed.
All our responses, point by point, are in red.
1. It is important to keep the review focused on GDM only- the terms T2DM, DM and GDM have been loosely used in various places of the text with no good integration
Thank you. We fully agree as the lines separating them get unclear. We have replaced DM with T2DM as over 99% of DM is T2 DM. We have also corrected portions of the manuscript that mentioned diabetes. In fact, we have revised major portions of the manuscript. Thank you.
2. For example, the term 'DM2'- line 84 is misleading-'DM'-line 124- please revise and be consisten
In the revised manuscript, we have been consistent throughout the manuscript. Thank you.
3. The content of the review is not well organized- is this simply a review of prevalence data or the causes and potential management strategies?
We have tried to show that there is little data available. At bare minimum, even the prevalence is not available. Since management of GDM is available from international organizations, countries would benefit from them. The International Federation of Gynecology and Obstetrics (FIGO), agrees that management can be modified to local circumstances: diagnosis, treatment and follow-up. We were hoping to get follow-up data, however, there is a big paucity –almost uniformly. We have stressed that following these women with GDM could help to contain the epidemic of DM. We have expanded and made major changes.
4. subhead number 5- reason for high prevalence of DM and GDM..- this section is very poorly written.
We appreciate your constructive criticism. We have paraphrased major portions of this subheading.
5. A comparative table on country wise prevalence and etiologic factors will be very helpful
We list the prevalence country-wise. However, few studies list the etiological factors for GDM. Most etiological factors are well known, and a few studies mention them so we just mention them in the manuscript text. This is true for almost all the Arab countries. The occasional ones that mention etiologic factors, we have elaborated in the manuscript, e.g. 7.4 Qatar.
6. Overall, the review contributes little in this area
This review was to look at available data on GDM in the Arab countries. We had no plans to write a comprehensive paper on all aspects of GDM including management. We were hoping to look some aspects of GDM, like prevalence, criteria used and follow-up, but there was little data available: even countries with well-developed medical schools and diabetes organizations, had little data on GDM. There was data on DM, but a paucity of information on GDM. What we have tried to stress is that more public health measures are urgently needed to follow GDM: it may help to contain the high prevalence of diabetes. A minor effort on prevention is far better that major efforts on cure. We hope that the new rewritten manuscript appeals to you. Thank you for all your effort.
Reviewer 3 Report
The review titled: Gestational diabetes in the Arab Gulf countries: sitting on a land-mine
This review was well referenced and provided an interesting overview regarding surge in the trends of T2DM and GDM in the Arab countries.
Overall, the structure is clear, and the English is fine. However, there are lots of typos and some editing (spacing errors) that needs to be performed throughout the manuscript. For example: Table 1 the use of "Some" is not very precise or appropriate for this context. The author should fix some of the short/fragmented lines (Lines:316-317; 335 and Typos e.g. line 101).
In table 2, specify “in Arab countries” in the title. Perhaps writing “unknown” in table 2 instead of the use of a “?” would be better.
Within the Abstract the author states that this review will address “b) the failure of containing it in Arab nations”. This was not very clearly stated for the different countries, could this be made clearer?
What are the types of treatments typically recommended during a GDM diagnosis during the pregnancy (e.g. is the treatment with insulin/ or diet or nothing?) in each of these countries? It would be interesting to see how these differ across the Arab countries.
In the future directions section: the author is correct in highlighting the importance for following up on the mother and child. However, could the author elaborate more on what uniform screening methods and protocols they think would be ideal and realistic (as well as preventative measures).
In the countries where 75g data is missing is there perhaps additional 50g testing information available, that the author could use?
Author Response
Thank you for your time and generous review. All your help has given our manuscript a new horizon. We are very grateful, indeed.
All our responses, point by point, are in red.
The review titled: Gestational diabetes in the Arab Gulf countries: sitting on a land-mine
This review was well referenced and provided an interesting overview regarding surge in the trends of T2DM and GDM in the Arab countries.
Thank you for your efforts with our review. However, we have rewritten major portions of the manuscript.
Overall, the structure is clear, and the English is fine. However, there are lots of typos and some editing (spacing errors) that needs to be performed throughout the manuscript. For example: Table 1 the use of "Some" is not very precise or appropriate for this context. The author should fix some of the short/fragmented lines (Lines:316-317; 335 and Typos e.g. line 101).
We appreciate your help. We have worked on the manuscript extensively. All the typos and fragmented lines have been corrected. Thank you.
In table 2, specify “in Arab countries” in the title. Perhaps writing “unknown” in table 2 instead of the use of a “?” would be better.
We have made the change. Thank you.
Within the Abstract the author states that this review will address “b) the failure of containing it in Arab nations”. This was not very clearly stated for the different countries, could this be made clearer?
We have rewritten major portions of the abstract to satisfy all our reviewers. We are sure that you shall like this new abstract. Since almost all countries barely have any follow-up for GDM in their guidelines (if at all present), we have made these remarks in the final conclusion and future directions.
What are the types of treatments typically recommended during a GDM diagnosis during the pregnancy (e.g. is the treatment with insulin/ or diet or nothing?) in each of these countries? It would be interesting to see how these differ across the Arab countries.
This review was to find all the available data on GDM in the Arab countries. We had no plans to write a comprehensive paper on all aspects of GDM including management and treatment. We were hoping to look some aspects of GDM, like prevalence, criteria used and follow-up, but there was little data. Even countries with well-developed medical schools and diabetes organizations, had little data on GDM. There was data on DM, but little on GDM. What we have tried to stress is that more public health measures are urgently needed to follow GDM: it may help to contain the high prevalence of diabetes.
In the future directions section: the author is correct in highlighting the importance for following up on the mother and child. However, could the author elaborate more on what uniform screening methods and protocols they think would be ideal and realistic (as well as preventative measures).
We have added the International Federation of Gynecology and Obstetrics (FIGO) recommendation that depending on the funds available, every country can modify major recommendations for screening and diagnosis and follow-up of GDM. We have sub-divided this section into four subheadings. This is the most crucial part of our review. We hope you like it.
In the countries where 75g data is missing is there perhaps additional 50g testing information available, that the author could use?
Thank you for the suggestion. However, the 50-g being only an screening test, it is used as such being followed by the 75-g or 100-g OGTT. We have not encountered any country where it is used alone. Sure, it is better than nothing. The bigger problem was that some countries are assuming the GDM and T2DM are the same, with the same diagnostic criteria.
We appreciate all your comments. They have been constructive and very helpful. Thank you.
Round 2
Reviewer 1 Report
Thank you for your response to the initial review and for the opportunity to review your revised manuscript. The following comments, questions, and suggestions are offered with all due respect.
General question
In the response to the initial review, there is mention of "we." There is only one author listed. Are there other authors?
Abstract
Lines 15-17: The author may wish to rephrase for clarity. For example, "unhealthy changes in diet and lifestyle." Is diet part of lifestyle? (Line 68 refers to overconsumption.) (Lines 82-83 could be rephrase from "... lifestyle changes, decreased physical activity, and increased calorie intake..." to "lifestyle changes including decreased physical activity..." Another example, "The excess nutrition turned the advantageous genes, originally selected for famine-like conditions, detrimental..." Is the challenge the fact that there is "excess nutrition" or is it the overconsumption of foods, unhealthy food/beverage options, something else?
Introduction
Lines 29-32: The author may wish to rephrase for clarity.
Table 1: Instead of using parentheses, should these be commas?
2. The modern world's epidemic...
Line 70: The author may wish to reconsider the use of the word "debilitating." Managed well, T2DM, GDM, or any other type of diabetes is not necessarily "debilitating."
3. The relationship between GDM and Type 2...
Line 89-90: The author may wish to rephrase for clarity. "GDM was originally defined to predict GDM in pregnant women after delivery."
4. Diabetes in the Arab world
Lines 105-107: The author may wish to rephrase for clarity. It is unclear what "...while representing 29 diabetes organizations..." refers to.
Lines 111-112: "Close to half 111 (45.0%) of this population was undiagnosed with T2DM." Of which population? Does information in this paragraph come from reference 16? If not, it may help to add additional references.
Lines 115-120: The author may wish to rephrase this paragraph.
5. Reason for high prevalence...
The author may wish to rephrase this paragraph to minimize redundancy.
Line 148: Should "thrifty genes" read as "drifty genes"?
Line 152: "it was beneficial" What was beneficial?
6. Genetic hypotheses and T2DM
The author may consider omitting this section because it is not relevant to GDM.
7.1. Saudi Arabia
Lines 183-184: The author may consider using "women with diabetes" instead of "diabetic women"
7.3. Bahrain
The author may wish to rephrase for clarity: "If the newer IADPSG criteria were used, since they give a higher 221 prevalence..."
Line 239: The author may wish to rephrase for clarity
7.5. United Arab Emirates (UAE)
Lines 264-265: It may be helpful to focus on adults, thereby omitting the last sentence.
7.7. Lebanon
Line 283: It will be helpful to use terms consistently. For example, "diabetes mellitus" is used here. Is this referring to T2DM? If so, T2DM would be more appropriate. [Same for Line 290, 294]
7.12. Oman
Line 319: "glucose tolerance" Should this be "glucose intolerance"?
Line 343: "they are known to give a much higher prevalence." Is there a reference to support this?
8.1. Increasing awareness among care- receivers and care-givers
Line 383: The author may wish to use a different word than "ignorance" as this word has a pejorative connotation.
Author Response
Thank you for your response to the initial review and for the opportunity to review your revised manuscript. The following comments, questions, and suggestions are offered with all due respect.
General question
In the response to the initial review, there is mention of "we." There is only one author listed. Are there other authors?
There are no other authors. I have used ‘we’ as using ‘I’ seemed to be somewhat arrogant. Also, since most of my studies on GDM have multiple authors, I am used to using ‘we.’ This is one of my few solo reviews.
Abstract
Lines 15-17: The author may wish to rephrase for clarity. For example, "unhealthy changes in diet and lifestyle." Is diet part of lifestyle? (Line 68 refers to overconsumption.) (Lines 82-83 could be rephrase from "... lifestyle changes, decreased physical activity, and increased calorie intake..." to "lifestyle changes including decreased physical activity..." Another example, "The excess nutrition turned the advantageous genes, originally selected for famine-like conditions, detrimental..." Is the challenge the fact that there is "excess nutrition" or is it the overconsumption of foods, unhealthy food/beverage options, something else?
Thank you for all your observations. We have corrected all of them. We hope that you are happy with the corrected version.
Lines 15-17: We have rephrased it for clarity. We have specified that physical activity and diet are a part of lifestyle. The same changes are in lines 82-83 and similar for consistency. We have also clarified that the excess nutrition really is excess calories.
We appreciate your comments. Thank you.
Introduction
Lines 29-32: The author may wish to rephrase for clarity.
This is an excellent suggestion. We have rephrased the lines. Thank you.
Table 1: Instead of using parentheses, should these be commas?
We have used parenthesis and clarified that these guidelines are the same. Thus NICE (2008) in the UK adopted the WHO (1999) guidelines. We agree that it was confusing and now it is clear. Thank you.
- The modern world's epidemic...
Line 70: The author may wish to reconsider the use of the word "debilitating." Managed well, T2DM, GDM, or any other type of diabetes is not necessarily "debilitating."
We have removed debilitating. Thank you.
- The relationship between GDM and Type 2...
Line 89-90: The author may wish to rephrase for clarity. "GDM was originally defined to predict GDM in pregnant women after delivery."
We have paraphrased this sentence for clarity.
- Diabetes in the Arab world
Lines 105-107: The author may wish to rephrase for clarity. It is unclear what "...while representing 29 diabetes organizations..." refers to.
This is an excellent suggestion. We fully agree that it is unclear. Thank you for your in-depth reading.
Lines 111-112: "Close to half 111 (45.0%) of this population was undiagnosed with T2DM." Of which population? Does information in this paragraph come from reference 16? If not, it may help to add additional references.
Yes, this is from the reference 16. We have paraphrased for clarity. Thank you.
Lines 115-120: The author may wish to rephrase this paragraph.
We appreciate your help. This paragraph was illogical with its flow. We have resurrected it. Indeed, it is much better now. Thank you.
- Reason for high prevalence...
The author may wish to rephrase this paragraph to minimize redundancy.
We have removed all the redundancy and paraphrased this paragraph. Thank you.
Line 148: Should "thrifty genes" read as "drifty genes"?
We have elucidated in the revised version that drifty and thrifty are distinct. So, now there should be no doubt at all. We hope that you are happy with this new version.
Line 152: "it was beneficial" What was beneficial?
We qualify that it was beneficial for the survival of a population in adverse conditions.
- Genetic hypotheses and T2DM
The author may consider omitting this section because it is not relevant to GDM.
We agree that it was out of place. However, we have replaced it with “Genetic associations of GDM and T2DM as a cause of obesity amongst Arabs” We have rewritten major portions, cut a lot of flab related to T2DM and relate the common susceptibility genes to obesity and consanguinity.
Thank you for this comment. It has really improved the manuscript.
7.1. Saudi Arabia
Lines 183-184: The author may consider using "women with diabetes" instead of "diabetic women"
We have changed this adjective to noun. We also checked rest of the document for “diabetic women,” however, there were none. Many thanks.
7.3. Bahrain
The author may wish to rephrase for clarity: "If the newer IADPSG criteria were used, since they give a higher 221 prevalence..."
We have rephrased and made it clear. Thank you.
Line 239: The author may wish to rephrase for clarity
We have done so. Thank you.
7.5. United Arab Emirates (UAE)
Lines 264-265: It may be helpful to focus on adults, thereby omitting the last sentence.
We have rephrased this to explain why diabetes is spilling over even in children. Young girls becoming pregnant is another problem of T2DM in the young.
7.7. Lebanon
Line 283: It will be helpful to use terms consistently. For example, "diabetes mellitus" is used here. Is this referring to T2DM? If so, T2DM would be more appropriate. [Same for Line 290, 294]
We have changed this. Thank you.
7.12. Oman
Line 319: "glucose tolerance" Should this be "glucose intolerance"?
Thank you for picking this oversight.
Line 343: "they are known to give a much higher prevalence." Is there a reference to support this?
We have added a reference. Thank you.
8.1. Increasing awareness among care- receivers and care-givers
Line 383: The author may wish to use a different word than "ignorance" as this word has a pejorative connotation.
We apologize. We did not ever imply a condescending attitude. The tone should always be of humility. We have changed it. Thank you.
We appreciate all your help and your critical comments. These have given the manuscript a new dimension in logic and clarity. Thank you.
Reviewer 2 Report
The author has adequately addressed all requested revisions.
However, some sentences are still difficult to read and make sense of. Recommend moderate English language editing.
Author Response
The author has adequately addressed all requested revisions.
Thank you.
However, some sentences are still difficult to read and make sense of. Recommend moderate English language editing.
We have made more language changes as suggested by our reviewer 1. We have carefully searched the manuscript for language editing. We appreciate all your effort towards our manuscript. We hope you find the new version much improved. Thank you.
Round 3
Reviewer 1 Report
Thank you for the opportunity to review this version of your manuscript. The following comments and suggestions are offered with all due respect.
Line 59: It is unclear who "underlying population" is referring to.
Lines 59-63: The author may wish to further explain why the prevalence of T2DM is a "reasonable estimate" of GDM. The fact that T2DM is more clearly defined does not necessarily mean that it can serve as a proxy. It may help to establish this relationship with information from Lines 92-103.
Lines 85-87: This sentence may need to be reworded for clarity.
Line 144: Does the author mean "unmasking," not "unmaking"?
Lines 197-198: "the knowledge of pregnant women was poor? Or, knowledge about GDM by pregnant women was poor?
7.1, 7.3, 7.9: These sections include specific ways that screening, education, etc. can be improved. If possible, it would be helpful to include this in all sections. Or, to include it only in the future directions/discussion sections.
Lines 273-274: It may be helpful to note that some of the young girls with glucose intolerance are at-risk for T2DM and/or GDM in adulthood.
Lines 406-407: "... Arab women migrating to the western world where the GDM prevalence is lower, can influence the prevalence of the adapted country [70]." Is this due to genetic disposition?
Lines 409-410: It is unclear what "the effect migration" is referring to.
Line 417: Is "health care provider not seeing the patient" different than patient being lost to follow-up?
Line 447: Would the "landmark" include the use of diagnostic guidelines or streamlining the guidelines?
Author Response
ijerph-1012043; Gestational diabetes in the Arab Gulf countries: sitting on a land-mine
Reviewer 1:
Thank you for the opportunity to review this version of your manuscript. The following comments and suggestions are offered with all due respect.
Line 59: It is unclear who "underlying population" is referring to.
We have clarified this further. We hope you are satisfied with our new changes.
Lines 59-63: The author may wish to further explain why the prevalence of T2DM is a "reasonable estimate" of GDM. The fact that T2DM is more clearly defined does not necessarily mean that it can serve as a proxy. It may help to establish this relationship with information from Lines 92-103.
We appreciate your valid criticism. We have expanded both these sections. To the author’s mind, this association was taken for granted—and it showed in the manuscript. However, we have overcome this overture and explained at some length. Of course, you cannot use T2DM as proxy. All one can say is that if the T2DM prevalence is high, so must be of GDM. Our reference 14 elaborates in much greater detail. Thank you for this very important critical point.
Lines 85-87: This sentence may need to be reworded for clarity.
We have paraphrased this sentence. Thank you.
Line 144: Does the author mean "unmasking," not "unmaking"?
We apologize for this typo. We have corrected it. Thank you.
Lines 197-198: "the knowledge of pregnant women was poor? Or, knowledge about GDM by pregnant women was poor?
Thank you. We appreciate your comment as knowledge does not imply knowledge about GDM. We have clarified. Thank you.
7.1, 7.3, 7.9: These sections include specific ways that screening, education, etc. can be improved. If possible, it would be helpful to include this in all sections. Or, to include it only in the future directions/discussion sections.
This is a capital idea. We have expanded this idea in the future direction section.
Lines 273-274: It may be helpful to note that some of the young girls with glucose intolerance are at-risk for T2DM and/or GDM in adulthood.
We have made the requisite changes. Thank you.
Lines 406-407: "... Arab women migrating to the western world where the GDM prevalence is lower, can influence the prevalence of the adapted country [70]." Is this due to genetic disposition?
We were trying to imply that migration of a population of expatriates with low GDM prevalence to a high prevalence (e.g., European expatriates in Saudi Arabia) would affect the reported prevalence-and vice versa. We have clarified this nebulous area.
Lines 409-410: It is unclear what "the effect migration" is referring to.
We have changed the sub-heading and rewritten portions of this paragraph. Thank you.
Line 417: Is "health care provider not seeing the patient" different than patient being lost to follow-up?
We used our reference 71 as the guide. We can only guess that they mean the obstetric care team as the patients usually go to the baby-follow up clinics. We have clarified. Thank you.
Line 447: Would the "landmark" include the use of diagnostic guidelines or streamlining the guidelines?
Here we are implying follow-up. We have data on a follow-up study in the UAE, which we are nearly ready to submit. We knew follow-up would be poor, but the follow-up figures are disturbing and too low. We have clarified in the MS. Thank you.
Thank you for all your time and all your suggestions. We appreciate all your help.
This manuscript is a resubmission of an earlier submission. The following is a list of the peer review reports and author responses from that submission.